# The Role of Centralized Sexual Assault Care Centers in HIV Post-Exposure Prophylaxis Treatment Adherence: A Retrospective Single Center Analysis

**DOI:** 10.3390/idr17040077

**Published:** 2025-07-03

**Authors:** Stefano Malinverni, Shirine Kargar Samani, Christine Gilles, Agnès Libois, Floriane Bédoret

**Affiliations:** 1Emergency Department, Université Libre de Bruxelles, 1000 Brussels, Belgium; shirine.kargar.samani@ulb.be (S.K.S.); floriane.bedoret@gmail.com (F.B.); 2Gynecology Department, Université Libre de Bruxelles, 1000 Brussels, Belgium; christine.gilles@stpierre-bru.be; 3Infectious Diseases Department, Université Libre de Bruxelles, 1000 Brussels, Belgium; agnes.libois@stpierre-bru.be

**Keywords:** post-exposure prophylaxis, non-occupational post-exposure prophylaxis, sexual assault, sexual assault victim, sexually transmitted diseases

## Abstract

Background: Sexual assault victims involving penetration are at risk of contracting human immunodeficiency virus (HIV). Post-exposure prophylaxis (PEP) can effectively prevent HIV infection if initiated promptly within 72 h following exposure and adhered to for 28 days. Nonetheless, therapeutic adherence amongst sexual assault victims is low. Victim-centered care, provided by specially trained forensic nurses and midwives, may increase adherence. Methods: We conducted a retrospective case–control study to evaluate the impact of sexual assault center (SAC)—centered care on adherence to PEP compared to care received in the emergency department (ED). Data from January 2011 to February 2022 were reviewed. Multivariable logistic regression analysis was employed to determine the association between centralized specific care for sexual assault victims and completion of the 28-day PEP regimen. The secondary outcome assessed was provision of psychological support within 5 days following the assault. Results: We analyzed 856 patients of whom 403 (47.1%) received care at a specialized center for sexual assault victims. Attendance at the SAC, relative to the ED, was not associated with greater probability of PEP completion both in the unadjusted (52% vs. 50.6%; odds ratio [OR]: 1.06, 95% CI: 0.81 to 1.39; *p* = 0.666) and adjusted (OR: 0.81, 95%CI 0.58–1.11; *p* = 0.193) analysis. The care provided at the SAC was associated with improved early (42.7% vs. 21.5%; *p* < 0.001) and delayed (67.3% vs. 33.7%; *p* < 0.001) psychological support. Conclusions: SAC-centered care is not associated with an increase in PEP completion rates in sexual assault victims beyond the increase associated with improved access to early and delayed psychological support. Other measures to improve PEP completion rates should be developed. **What is already known on this topic**—Completion rates for HIV post-exposure prophylaxis (PEP) among victims of sexual assault are low. Specialized sexual assault centers, which provide comprehensive care and are distinct from emergency departments, have been suggested as a potential means of improving treatment adherence and completion rates. However, their actual impact on treatment completion remains unclear. **What this study adds**—This study found that HIV PEP completion rates in sexual assault victims were not significantly improved by centralized care in a specialized sexual assault center when compared to care initiated in the emergency department and continued within a sexually transmitted infection clinic. However, linkage to urgent psychological and psychiatric care was better in the specialized sexual assault center. How this study might affect research, practice or policy—Healthcare providers in sexual assault centers should be more aware of their critical role in promoting PEP adherence and improving completion rates. Policymakers should ensure that measures aimed at improving HIV PEP outcomes are implemented at all points of patient contact in these centers. Further research is needed to assess the cost-effectiveness of specialized sexual assault centers.

## 1. Introduction

The World Health Organization defines sexual violence as any sexual act, attempt to obtain a sexual act, or other act directed against a person’s sexuality using coercion, by any person regardless of their relationship to the victim, irrespective of the setting [1]. According to this definition, 64% of the Belgian population aged 16 to 69 has experienced at least one form of sexual violence [2]. Victims of sexual violence involving penetration are at risk of contracting sexually transmitted infections (STIs), including human immunodeficiency virus (HIV). Sexual assault increases the risks of HIV transmission due to both the increased risk of traumatic lesions to the epithelium and the prevalence of the perpetrator [3]. Post-exposure prophylaxis (PEP) is part of the comprehensive medical care provided to sexual assault victims to mitigate the risk of HIV infection [4]. For optimal efficacy, PEP must be initiated as soon as possible, within 72 h following exposure [5].

Completion rates amongst sexual assault victims are low, averaging 40% globally and 33% in developed countries [6,7]. These low rates are alarming, as the effectiveness of PEP is related to completion of a 28-day course of treatment [8]. Multiple barriers to adherence exist, including treatment side effects, stigma, emotional distress, forgetfulness or treatment costs [9,10]. Following sexual violence, victims require urgent medical attention for STI risk assessment and to determine the necessity of PEP, along with psychological, forensic and comprehensive medical care. Initial care settings vary across countries and regions with differing HIV prevalence. Belgium is a low prevalence country where prevalence outside high-risk groups is estimated below 0.02%. In Belgium, HIV PEP for sexual assault victims has traditionally been administered in emergency departments (EDs), guided by national guidelines to decide on treatment initiation according to the exposure and delivered by personnel often lacking specialized training in sexual assault care [11].

Victim-centered care, delivered by specially trained forensic nurses and midwives, has emerged as an alternative to improve the quality of medical and psychological care [12,13,14]. Consequently, ten Sexual Assault Centers (SACs) have been established in Belgium since 2017, aiming to provide high-quality care to sexual assault victims and improve PEP adherence. Victims have a 24/7 access to specialized medical and forensic care, psychological assistance and longitudinal integrated care within a safe environment [15]. Our hospital has hosted the first Belgian SAC since January 2018.

Although results from previous studies suggest that sexual assault centers could improve PEP, evidence is scarce and based on small cohorts, supporting the need for a larger studies assessing SAC impact on PEP adherence. To fill this knowledge gap, we conducted a retrospective observational cohort study to assess the impact of specialized holistic care provided in a SAC on PEP completion and psychological support, compared with initial care provided at the ED. For our primary analysis, we hypothesized that SAC would improve PEP completion rates.

## 2. Methods

### 2.1. Study Design and Participants

This observational retrospective study utilized prospectively collected data from the Centre Hospitalier Saint-Pierre, a tertiary university hospital housing an HIV reference center and providing specialized care for sexual and gender minorities, located in an underprivileged, high-immigration area of Brussels, Belgium. Between January 2011 and February 2022, we recorded data on sexual assault victims consulting for PEP following suspected or confirmed sexual assault at our HIV reference center. Prior to December 2017, sexual assault victims were assessed for PEP after potential HIV exposure by a gynecologist or a surgeon at the ED according to the national guidelines [11]. Since January 2018, sexual assault victims receive care at the SAC where specialized, multidisciplinary, and longitudinal integrated care is provided by forensic nurses and midwives supported by gynecologists and infectious diseases specialists.

Our research was conducted in accordance with the Declaration of Helsinki and local legislation. The study protocol was approved by our institutional review board (CE/22-04-08, approved on 7 April 2022) and patient consent was waived as per study protocol given the retrospective observational design.

### 2.2. Care Pathways

Before the opening of the SAC, sexual assault victims were managed by a gynecologist or a surgeon in the ED and received a 5-day PEP starter pack to bridge them to their first outpatient consultation in the HIV reference center. Oral and written instructions on the follow-up clinic appointment and the PEP medication were provided to all participants. Immediate psychological support was offered in the ED (limited to working hours), with support deferred to follow-up consultations. Follow-up visits were scheduled within 5 days and at 2, 6, and 12 weeks post-assault. During outpatient consultations with an infectious disease specialist, the exposure-associated risk was re-evaluated, side effects were assessed, STI counseling and screening were offered, further follow-up testing was planned according to exposure characteristics and a decision regarding the continuation or cessation of PEP were made.

Victims initially cared for in the SAC received care in a dedicated environment staffed by specialized multidisciplinary personnel available 24/7. They received a 7-day PEP starter pack with oral and written instructions regarding their follow-up SAC appointment and PEP medication. The follow-up schedule remained the same but included systematic weekly telephone follow-ups for two months to evaluate victims’ mental and physical health status. All their follow-up care was delivered within the SAC from the specialized multidisciplinary staff. Optional follow-up with a psychologist specialized in trauma related to sexual assault was available upon patient request. Immediate care, follow-up visits, and medications are reimbursed for patients cared for at the SAC, contrasting to the exclusively immediate assistance and medications prior to its establishment.

### 2.3. Data Collection

Data were extracted from PEP-specific electronic health records by an independent statistician not involved in the study, using a standardized data collection form. Recorded variables were age, sex, sexual orientation, health insurance status, comorbidities, sexual-assault-specific variables (type of intercourse, number of aggressors, presence of ejaculation, presence of mucosal tears or infections), socio-demographic characteristics of the assailant, baseline blood tests, treatment, and observed adherence. We used electronic reports from the hospital pharmacy to cross-check treatment adherence in case of missing values in reported adherence in hospital records. Single patient data review was performed in case of incongruences in the data. Missing data were handled as missing and no imputation was realized.

### 2.4. Laboratory Procedures

Routine blood tests and serologies were performed at the emergency department or SAC according to protocolized care. Biological samples were collected for forensic investigations according to legal recommendations.

### 2.5. Outcome Measures

The primary outcome of the study was the completion of a 28-day course of PEP, as defined by both self-report and pharmacy records. Patients were considered compliant with their 28-day course if a chart review confirmed that they had completed the full 28 days according to self-report and as documented in their pharmacy records. Incomplete adherence with the PEP regimen was designated if the pharmacy chart or self-report indicated completion of fewer than 26 days of treatment. Patients whose treatment was prematurely discontinued based on physician advice or who expressed intent to complete their follow-up elsewhere were excluded from the analysis. The secondary outcome of the study was attendance at the first psychological support follow-up appointment within 5 days either at the HIV reference center or SAC.

Health insurance coverage was defined as the presence of either national public insurance or private insurance that covers medical care costs. Minors were defined as patients younger than 18 years. Patients were considered of Belgian origin if they were born in Belgium. The initial treatment was classified as single-tablet treatment (STT) if the prescribed treatment involved single fixed-dose tablet combining three antiretrovirals allowing treatment with one tablet a day.

Regular work hours were defined as Monday to Friday, 08:00 to 17:00 h while after-hours were defined as 17:00 to 08:00 h or a weekend day.

### 2.6. Statistical Analysis

On the basis of epidemiological data from a previous study [16], we calculated that with a sample size of 768 and an equal number of study participants in each group would have a power of 80% to detect a 10% absolute increase in terms of adherence of the SAC-cared group compared with the standard, ED-based-cared group. As we included 856 patients in the analysis with similar numbers of patients in each group, we considered the sample size to be adequate and proceeded with the analysis.

Variables are reported as medians with interquartile ranges (IQRs) if not Gaussian or as means ± standard deviations if they were normally distributed. Comparisons were assessed by using the Mann–Whitney U test for continuous variables, whereas categorical variables were assessed by χ^2^ test.

For the primary and secondary outcomes, multivariable logistic regression models were used to estimate the between-group difference in the proportion of patients terminating the PEP regimen as well as the difference in the proportion of patients who attended the first follow-up consultation following the sexual assault. Confounders for the multivariable models were selected according to an historical approach [16,17] as recommended for causal inference [18]. Our model controlled for age, sexual orientation, migratory status, knowledge of the aggressor, STT, consultation during office hours and administration of psychological support within 5 days. ORs and mean differences were reported with 95% confidence intervals (CIs). We omitted side effects from the univariate and multivariable analysis as we observed a significant reporting bias in patients retained in care. Statistical significance was defined as a two-sided *p* value of less than 0.05. All statistics were performed with Stata, version 16 (Statacorp, College Station, TX, USA).

## 3. Results

During the study period, 5697 patients consulted following a sexual assault. Most of these patients were deemed ineligible for PEP mostly due to delayed presentation or lack of evidence of penetration. A total of 43 patients declined PEP, while 86 indicated they would pursue follow-up care elsewhere and 46 interrupted PEP following physician advice. This resulted in 856 patients available for the analysis (Figure 1), 403 receiving care initially at SAC and 453 at the ED (52.9% ED, 47.1% SAC).

The majority of patients (89.8%) were female and there were no transgender patients. Median age was 26 years (IQR: 21–33). The health insurance coverage was 90% overall, and it was significantly higher in the SAC group than in the ED group (98.5% vs. 80.4%; *p* < 0.001). The median time to presentation was higher in patients cared at the SAC (17 [IQR: 7–36] h versus [vs.] 15 [7,8,9,10,11,12,13,14,15,16,17,18,19,20,21,22,23,24,25,26,27,28,29]; *p* < 0.001). 68.8% of the victims consulted outside office hours. Elvitegravir/Cobicistat/Emtricitabine/Tenofovir (EVG/COBI/FTC/TDF)-based STT was more frequent in patients cared by the SAC (79.2% vs. 46.4%, *p* < 0.001) (Table 1).

Among all victims, 75% reported that the sexual assault included vaginal receptive intercourse with a lower proportion within patients cared by the SAC (66.7% vs. 84.6%; *p* < 0.001), 27.9% reported anal receptive intercourse and 26.8% reported oral receptive intercourse. The aggression was perpetrated by more than one aggressor in a lower proportion of patients seeking care at the SAC (12.4% vs. 20.1%; *p* = 0.002). A higher proportion of patients seeking care at the SAC knew the aggressor (46.6% vs. 25.1%; *p* < 0.001) (Table 2).

Victims attended at the SAC had similar chances to complete the 28-day PEP course with an observed completion rate of 53.6% compared with 46.4% for patients initially cared in the ED (OR:1.06, 95% CI: 0.81 to 1.39, *p* = 0.666) (Table 3).

### 3.1. Multivariable Regression

After adjustment for known confounders PEP treatment provided within the SAC was not associated with higher treatment completion rates (OR: 0.81, 95% CI: 0.58 to 1.11, *p* = 0.193) (Table 3). Independent predictors of treatment completion were homosexual sexual orientation (OR: 2.74, 95% CI: 1.49 to 5.05, *p* = 0.001), health insurance (OR: 2.05, 95% CI: 1.22 to 3.45, *p* = 0.007), early psychological support (OR: 2.02, 95% CI: 1.45 to 2.79, *p* < 0.001), STT (OR: 1.57, 95% CI: 1.14 to 2.17, *p* = 0.006), and assault from multiple aggressors (OR: 0.66, 95% CI: 0.44 to 0.98, *p* = 0.040).

### 3.2. Secondary Outcome

Psychological support was provided to 278 patients (32.8%) within five days from the assault with a higher proportion in the SAC group compared to the control group (42.7% vs. 21.5%; *p* < 0.001) (Table 4).

After adjustment for known confounders, psychological support care at the SAC was significantly associated with access to early psychological care (OR: 2.93, 95% CI: 2.09–4.1, *p* < 0.001).

## 4. Discussion

In this retrospective single-center study, SAC-centered care was not associated with increased completion rates of the prescribed 28-day PEP, regardless of observed patient and assault characteristics. However, early psychological support was higher and independently associated with care provided in the SAC compared to patients treated in the ED.

Our analysis revealed notable differences between patients treated before and after the establishment of the SAC. Patients initially seen in the ED were younger and more often migrants, assaults more frequently involved receptive vaginal penetrations and multiple assailants. Conversely, patients presenting at the SAC were more likely to know their assailant, possess health insurance, receive single-tablet treatment and consult a psychologist within five days of initiating treatment. Therefore, characteristics associated with greater treatment adherence were more prevalent among the SAC population [16,17,19]. Following the opening of the SAC, adherence rates increased slightly from 50.6% to 52.1%, without a statistically significant difference in multivariate analysis.

The literature suggests that patient-centered care focusing on physical and emotional well-being of the patient, provided by trained professionals, should enhance initial care and follow-up, thereby improving PEP adherence [20,21]. The SAC provides high-quality care by qualified staff whereas physicians working in the ED often lack specialized training in this area [22]. Our results deviate from previous findings and several factors may account for the lack of improvement in treatment adherence following the establishment of the SAC.

First, we observed that victims seeking care at the SAC were significantly more likely to know their assailant—defined as an acquaintance or partner—compared to those initially treated in the ED (46.6% vs. 25.1%, *p* = 0.000). This higher proportion may be related to the fact that accessing care at the SAC does not require police involvement, whereas victims presenting to the ED were often accompanied by law enforcement following a formal report. Notably, knowing the assailant has been associated with a higher likelihood of non-reporting [23], lower rates of treatment initiation and increased rates of PEP non-completion [24].

Second, we observed that 25% of patients at the SAC initiated treatment, compared to only 11% in the ED. While higher initiation may reflect better access, it also increases the number at risk of discontinuation. We hypothesize that patients attended at the SAC may feel pressured to begin treatment, potentially leading to PEP initiation among individuals with lower intrinsic motivation.

Third, while patients received weekly telephone follow-ups by a case manager, conversations predominantly focused on psychological well-being rather than treatment adherence. Consequently, the lack of reminders regarding PEP adherence during these follow-up calls may partially explain the absence of improved compliance among SAC patients Follow-up calls should incorporate reminders about the importance of treatment adherence and help identify potential PEP side effects to enhance adherence [25]. Such reminders are fundamental given that sexual assault victims may experience post-traumatic stress disorder, which can manifest with symptoms that interfere with medication adherence [26]. Symptoms only decrease after the first few months following the event [27], which could result in discontinuity in prophylaxis intake.

Fourth, previously reported improvements in PEP adherence and victim-centered care at SACs may be attributable to co-interventions such as STT and enhanced psychological support. After adjusting for these known confounders in our multivariable regression model, we found no independent effect associated with the SAC itself.

Additionally, the SARS-CoV-2 pandemic may have influenced follow-up care and treatment adherence during the SAC study period. Follow-up visits were disrupted during the pandemic, particularly during lockdown periods, potentially negatively impacting adherence as observed in other conditions [28].

Finally, the centralization of care associated with the SAC might adversely affect compliance. While victims initially treated in the ED were referred to various hospital services for subsequent appointments, patients attended at the SAC returned to the same location for follow-up visits. This can evoke memories of the traumatic experience at each appointment. Furthermore, the SAC’s designation may increase stigma, which can additionally inhibit adherence to non-occupational post-exposure prophylaxis PEP and perpetuate social isolation [29].

The SAC itself provides free access to resources that can help ensure optimal care and improve treatment adherence [30]. We observed that psychological support significantly improved, and this directly contributes to an increase in nPEP adherence. However, other factors affect adherence, such as the impact of trauma on patients, the pressure on treatment initiation and the type and quality of follow-up.

Current evidence on SAC primarily focuses on legal outcomes, victim experience and access to care; however, studies evaluating their impact on PEP adherence are lacking, underscoring the relevance of our findings.

Given the high-resource, low-prevalence context of our study, our findings are relevant to clinicians and policymakers, as they underscore the importance of ensuring that sexual assault centers intervene at every patient contact to enhance PEP adherence and completion rates. Reminders emphasizing the importance of PEP adherence should be provided systematically at each follow-up with careful consideration given to their mode of delivery. Linkage to psychological care improved but remains insufficient. Further efforts are needed to enhance access, as early psychological support in our study was associated with a two-fold increase in PEP completion. Embedding psychological care within the initial medical encounter may promote adherence and potentially reduce post-traumatic symptoms [31]. Policymakers should support the implementation of systematic PEP adherence reminders and ensure that linkage to psychological care is offered at every consultation.

Our study has certain limitations. Firstly, its retrospective nature may introduce biases related to the quality and completeness of data derived from electronic medical records. Secondly, as a before-and-after study, unobserved factors may have influenced treatment adherence over time. Thirdly, the two patient groups were not balanced in their observed characteristics, although we controlled for these differences in multivariable regression. Fourthly, the absence of a phase-out period and the immediate post-opening evaluation of the SAC do not account for a potential learning curve among healthcare providers. Finally, the single-center design limits the generalizability of the findings to other healthcare systems and differently organized SACs. The low HIV prevalence within our context might have influenced PEP completion rates. Further research is warranted to assess whether a multidisciplinary, cost-free, specialized, and centralized organizational model could improve PEP completion rate among victims of sexual violence when providing tailored interventions at every patient contact highlighting the importance of PEP adherence. The potential long term benefit of the observed increase in urgent psychological follow up has not been assessed in the current study and warrants further research.

In conclusion, a holistic PEP program for sexual assault victims should focus on adherence to the prescribed regimen and interventions improving PEP outcomes. Our study shows that instauration of a SAC does not improve completion rates of the prescribed 28 days PEP regimen while it improves access to early psychological support, which in turn increases treatment adherence.

## Figures and Tables

**Figure 1 idr-17-00077-f001:**
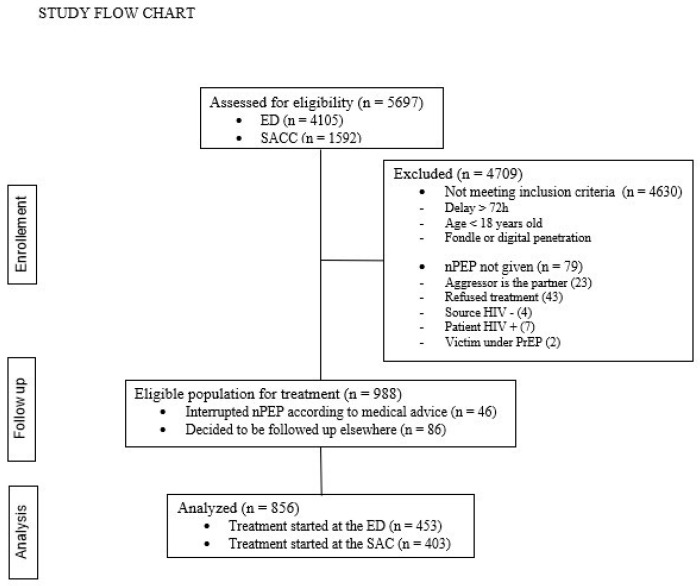
Represents the inclusion flow chart of the study detailing the participants selection.

**Table 1 idr-17-00077-t001:** Patients characteristics at study inclusion.

	Total Study Population	ED	SAC	*p* Value
	(No. = 856)	(No. = 403)	(No. = 453)	
**Median age, years (IQR)**	25.5 (21–33)	24 (20–32)	26 (21–34)	0.000
**Female, No. (%)**	769 (89.8)	370 (91.8)	399 (88.1)	0.071
**Sexual orientation, No. (%)**				0.369
**Homosexual**	61 (7.2)	24 (6.0)	37 (8.4)	
**Heterosexual**	745 (88.6)	362 (90.0)	383 (87.2)	
**Bisexual**	35 (4.2)	16 (4.0)	19 (4.3)	
**Migrant, No. (%)**	418 (48.9)	214 (53.1)	204 (45.2)	0.022
**Health insurance, No. (%)**	731 (90.0)	324 (80.4)	407 (98.5)	0.000
**Time to presentation, hours (IQR)**	16 (7–34)	15 (7–29)	17 (7–36)	0.000
**Office hours, No. (%)**	267 (31.2)	104 (25.9)	163 (36.0)	0.428
**Treatment, No. (%)**				0.000
**STT (EVG/COBI/FTC/TDF)**	546 (63.8)	187 (46.4)	359 (79.2)	
**LPV/d4T/3TC**	216 (25.2)	216 (53.6)	0 (0)	
**TDF/FTC/RAL**	94 (11)	0 (0)	94 (20.8)	

Abbreviations: EVG/COBI/FTC/TDF, Elvitegravir/Cobicistat/Emtricitabine/Tenofovir; IQR, inter-quartile range; LPV/d4T/3TC, Ritonavir-boosted Lopinavir/stavudine/lamivudine; No., number; STT, single tablet treatment; TDF/FTC/RAL, Tenofovir/Emtricitabine/Raltegravir.

**Table 2 idr-17-00077-t002:** Type of exposure.

	Total Study Population	ED	SAC	*p* Value
**Receptive anal, No. (%)**	240 (27.9%)	115 (28.5%)	125 (27.6%)	0.759
**Receptive vaginal, No. (%)**	643 (75.1%)	341 (84.6%)	302 (66.7%)	0.000
**Receptive oral, No. (%)**	229 (26.8%)	106 (26.3%)	123 (27.2%)	0.769
**Multiple aggressors, No. (%)**	137 (16%)	81 (20.1%)	56 (12.4%)	0.002
**Knowing the aggressor, No. (%)**	312 (36.4)	101 (25.1)	211 (46.6)	0.000

Abbreviations: No., number.

**Table 3 idr-17-00077-t003:** Unadjusted and adjusted determinants of completion of a 28-day PEP regimen.

	Adherence		Univariate Analysis		Multivariate Analysis	
	Yes (No. = 440)	No (No. = 416)	OR (95% CI)	*p* Value	OR (95% CI)	*p* Value
**Median age, age (IQR)**	25 (21–32)	26 (21–34)	0.99 (0.98–1.00)	0.110	0.99 (0.97–1.00)	0.067
**Male, No. (%)**	43 (9.95)	18 (4.40)	1.46 (0.93–2.29)	0.099		
**Homosexual, No. (%)**	62 (14.35)	34 (8.31)	2.42 (1.37–4.28)	0.002	2.74 (1.49–5.05)	0.001
**Bisexual, No. (%)**	19 (4.40)	16 (3.91)	1.20 (0.61–2.38)	0.594	1.11 (0.54–2.27)	0.783
**SAC, No. (%)**	236 (53.64)	217 (52.16)	1.06 (0.81–1.39)	0.666	0.81 (0.58–1.11)	0.193
**Health insurance, No. (%)**	395 (92.94)	336 (85.93)	2.16 (1.35–3.44)	0.001	2.05 (1.22–3.45)	0.007
**Migrant, No. (%)**	200 (45.66)	218 (52.40)	0.76 (0.58–1.00)	0.049	0.80 (0.59–1.08)	0.140
**Multiple aggressors, No. (%)**	57 (12.95)	80 (19.23)	0.63 (0.43–0.90)	0.012	0.66 (0.44–0.98)	0.040
**Amnesia, No. (%)**	198 (45.41)	161 (39.17)	1.29 (0.98–1.70)	0.066	1.28 (0.95–1.73)	0.111
**Knowing the aggressor, No (%)**	159 (36.14)	153 (36.78)	0.97 (0.74–1.28)	0.845	1.06 (0.90–1.26)	0.469
**Office hours, No. (%)**	147 (33.49)	120 (28.85)	1.24 (0.93–1.66)	0.144	1.22 (0.89–1.67)	0.224
**Psychological support within 5 days, No. (%)**	171 (39.13)	107 (26.03)	1.83 (1.36–2.45)	0.000	2.02 (1.45–2.79)	0.000
**Psychological support within 28 days, No. (%)**	278 (64.20)	158 (37.98)	2.93 (2.22–3.87)	0.000		
**STT, No. (%)**	303 (68. 86)	243 (58.41)	1.57 (1.19–2.08)	0.002	1.57 (1.14–2.17)	0.006

Abbreviations: IQR, inter-quartile range; No., number; STT, single tablet treatment.

**Table 4 idr-17-00077-t004:** Psychological support.

	Total Study Population	ED	SAC	*p* Value
	(No. = 856)	(No. = 403)	(No. = 453)	
**Within 5 days, No. (%)**	278 (32.8)	85 (21.5)	193 (42.7)	0.000
**Within 28 days, No. (%)**	436 (51.4)	136 (33.7)	300 (67.3)	0.000

Abbreviations: No., number.

## Data Availability

Data are available upon reasonable request and appropriate quoting.

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
