# Peer review of "The Role of Centralized Sexual Assault Care Centers in HIV Post-Exposure Prophylaxis Treatment Adherence: A Retrospective Single Center Analysis"

_2036-7449, 2025, doi:10.3390/idr17040077_

Round 1
Reviewer 1 Report
Comments and Suggestions for Authors
In the current article, the study discovered that the completion rates for HIV post-exposure prophylaxis (PEP) among sexual assault victims did not show significant improvement when care was centralized at a specialized sexual assault center, compared to care that began in the emergency department and continued at a sexually transmitted infection clinic. Also the noted point here, specialized sexual assault center did provide better access to urgent psychological and psychiatric care. The scope of the work is quite limited, making it more appropriate for specialized journals. Given that the study does not present a significant breakthrough and has a narrow range of novelty, it does not meet the journal's criteria for publication. As a result, it is not recommended for inclusion in the journal.

Author Response
We sincerely thank you for your thorough review of our manuscript entitled "The role of centralized sexual assault care centers in HIV post exposure prophylaxis treatment adherence. A retrospective single center analysis." We appreciate the insightful comments and constructive suggestions, which have helped us improve the quality and clarity of our work.
Below, we provide a detailed point-by-point response to each of the reviewers’ comments. Reviewer comments are presented in bold, followed by our responses in regular text with the modification to the manuscript highlighted in italic. All changes made in the manuscript are highlighted accordingly.
Reviewer 1
In the current article, the study discovered that the completion rates for HIV post-exposure prophylaxis (PEP) among sexual assault victims did not show significant improvement when care was centralized at a specialized sexual assault center, compared to care that began in the emergency department and continued at a sexually transmitted infection clinic. Also the noted point here, specialized sexual assault center did provide better access to urgent psychological and psychiatric care. The scope of the work is quite limited, making it more appropriate for specialized journals. Given that the study does not present a significant breakthrough and has a narrow range of novelty, it does not meet the journal's criteria for publication. As a result, it is not recommended for inclusion in the journal.
We thanks the reviewer for his/her comment on our work.
We would like to highlight a few aspects that might help to clarify why we kindly disagree with his/her assessment.
First, negative results in research are as important as positive results.
Our research indeed proves that completion rates of PEP for sexual assault victims is not improved by care provided in specialized centers, beyond the improvement provided by the better and earlier access to psychological support.
It does therefore provides two important insights to policy-makers and clinicians designing nPEP care for sexual assault victims:
- Psychological support is paramount and its access improves significantly nPEP completion rates
- Specialized centers do not improve per se completion rates. More tailored approaches need to be implemented within the specialized centers to tackle low nPEP completion rates.
Moreover, no previous research was performed to assess the impact of SAC on PEP completion rates
We hope that the revisions and clarifications provided address all concerns raised. We are grateful for the opportunity to revise our manuscript and look forward to your feedback.
Sincerely,
Stefano Malinverni, on behalf of all co-authors
Reviewer 2 Report
Comments and Suggestions for Authors
Overall, the paper is well written and organized
Line 12 can you specify the abbreviation SAC since that is the first time it’s mentioned in this paper
Line 13 contradicts lines 79 and 80, was the study period 2011 to 2022 or 2012 to 2022, please clarify
Line 139 and 140, is it private or public or private of public? kindly correct
The visual representation of figure 1 is very commendable, it’s clear to understand how research participants were included and excluded logically
Line 182 what does these acronym mean- EVG/COBI/FTC/TDF, they were first mentioned on line 182 so the meaning should be moved from line 194 to 182
Line 200 What were the known confounders? can you list some or all if possible
Line 200 is it SACC or SAC kindly clarify and correct
Author Response
We sincerely thank you for your thorough review of our manuscript entitled "The role of centralized sexual assault care centers in HIV post exposure prophylaxis treatment adherence. A retrospective single center analysis." We appreciate the insightful comments and constructive suggestions, which have helped us improve the quality and clarity of our work.
Below, we provide a detailed point-by-point response to each of the reviewers’ comments. Reviewer comments are presented in bold, followed by our responses in regular text with the modification to the manuscript highlighted in italic. All changes made in the manuscript are highlighted accordingly.
Line 12 can you specify the abbreviation SAC since that is the first time it’s mentioned in this paper
We provided the specification for the abbreviation SAC
Line 13 contradicts lines 79 and 80, was the study period 2011 to 2022 or 2012 to 2022, please clarify
We thank you for spotting the contradiction. It was a typo and we corrected it to 2011 to 2022.
Line 139 and 140, is it private or public or private of public? kindly correct
Thank you for the suggestion, we corrected it to: Health insurance coverage was defined as the presence of either national public insurance or private insurance that covers medical care costs.
By this we wanted to highlight that we defined as having an health insurance any patient for whom any additional cost would have been covered by an insurance.
The visual representation of figure 1 is very commendable, it’s clear to understand how research participants were included and excluded logically
We thank you for the positive comment. Indeed we followed the PRISMA guidelines for inclusion flowcharts.
Line 182 what does these acronym mean- EVG/COBI/FTC/TDF, they were first mentioned on line 182 so the meaning should be moved from line 194 to 182
We thank you for the suggestion and we added the acronym meaning to line 182. We left the meaning on line 194 so that the table can be understood as a stand-alone element.
Line 200 What were the known confounders? can you list some or all if possible
We thank you for the suggestion. We specified known confounders in the Statistical analysis paragraph by adding: Our model controlled for age, sexual orientation, migratory status, knowledge of the aggressor, STT, consultation during office hours and administration of psychological sup-port within 5 days.
Line 200 is it SACC or SAC kindly clarify and correct
Thank you for the suggestion. We corrected the typo.
We hope that the revisions and clarifications provided address all concerns raised. We are grateful for the opportunity to revise our manuscript and look forward to your feedback.
Sincerely,
Stefano Malinverni, on behalf of all co-authors
Reviewer 3 Report
Comments and Suggestions for Authors
1. Introduction:
Expand the contextualisation of HIV and sexual violence.
Include more previous studies on adherence to PEP and victim-centred care.
Explain the knowledge gap and justify the need for the study.
Clearly present the hypothesis.
2. Discussion:
Explore why SAC did not increase adherence to PEP.
Emphasise the positive impact of psychological support.
Compare with other similar studies.
Point out the limitations of the study and suggest areas for future research.
Finally, discuss the implications for public policy on victim care.
Comments on the Quality of English LanguageImprove your English.
Author Response
We sincerely thank you for your thorough review of our manuscript entitled "The role of centralized sexual assault care centers in HIV post exposure prophylaxis treatment adherence. A retrospective single center analysis." We appreciate the insightful comments and constructive suggestions, which have helped us improve the quality and clarity of our work.
Below, we provide a detailed point-by-point response to each of the reviewers’ comments. Reviewer comments are presented in bold, followed by our responses in regular text with the modification to the manuscript highlighted in italic. All changes made in the manuscript are highlighted accordingly.
Reviewer 3
We would like to thanks the Reviewer for the time taken and the feedback provided which, provided some suggestions improve the quality and clarity of the manuscript.
- Introduction:
Expand the contextualisation of HIV and sexual violence.
We added the following sentence and cited it appropriately to provide more context:
Sexual assault increases the risks of HIV transmission due to both the increased risk of traumatic lesions to the epithelium and the prevalence of the perpetrator
Include more previous studies on adherence to PEP and victim-centred care.
We thank you for the suggestion and we provided more references to previous studies on PEP adherence and victim centered care
Explain the knowledge gap and justify the need for the study.
Clearly present the hypothesis.
We thank you for the two above suggestions and we added the following two sentences (and adapted the sentence between the two) to clearly present the knowledge gap, the need for the study and our hypothesis.
Although results from previous studies suggest that sexual assault centers could im-prove PEP, evidence is scarce and based on small cohorts, supporting the need for a larger studies assessing SAC impact on PEP adherence. To fill this knowledge gap we conducted a retrospective observational cohort study to assess the impact of specialized holistic care provided in a SAC on PEP completion and psychological support, compared with initial care provided at the ED. For our primary analysis, we hypothesized that SAC would improve PEP completion rates.
- Discussion:
Explore why SAC did not increase adherence to PEP.
To explain why SAC did not increase adherence to PEP would be, we think, inappropriate as we don not have data supporting the mechanism why it did not work in our context. We could only emit some hypothesis such as the idea the SAC implementation in the previous literature came with measures such as STT and increased psychological support. As we corrected for these two mechanism (as known confounders) in our multivariable regression model we could not find any effect associated with the SAC per se. We already provided a long explanation exploring why the SAC did not increased adherence to PEP (lines 249 – 287) We modified our discussion and added the following sentences to explore this observation.
Fourth, previously reported improvements in PEP adherence and victim-centered care at SACs may be attributable to co-interventions such as STT and enhanced psychological support. After adjusting for these known confounders in our multivariable regression model, we found no independent effect associated with the SAC itself.
Emphasise the positive impact of psychological support.
We thank you for the comment.
We changed and expanded the paragraph mentioning the positive impact of psychological as follows:
Linkage to psychological care improved but remains insufficient. Further efforts are needed to enhance access, as early psychological support in our study was associated with a two-fold increase in PEP completion. Embedding psychological care within the initial medical encounter may promote adherence and potentially reduce post-traumatic symptoms.
Compare with other similar studies.
To our knowledge there are no studies on the impact of SAC on PEP adherence.
We added the following sentence:
Current evidence on SAC primarily focuses on legal outcomes, victim experience and access to care, however, studies evaluating their impact on PEP adherence are lacking, underscoring the relevance of our findings.
Point out the limitations of the study and suggest areas for future research.
We thank you for this suggestion.
A detailed list of limitations and future research suggestions is provided line 314-327
Finally, discuss the implications for public policy on victim care.
We thank you for this suggestion and added the following sentence:
Policymakers should support the implementation of systematic PEP adherence reminders and ensure that linkage to psychological care is offered at every consultation.
We hope that the revisions and clarifications provided address all concerns raised. We are grateful for the opportunity to revise our manuscript and look forward to your feedback.
Sincerely,
Stefano Malinverni, on behalf of all co-authors
Reviewer 4 Report
Comments and Suggestions for Authors
The authors analyse the completion rate of PEP after rape depending on centre of treatment in Belgium. No differences were found between emergency rooms and specialized care providers. However, psychological care seemed to be better in specialized centres.
The study is well conducted and the manuscript well written. The introduction is short but concise and well organized. The study question is well explained at the end of the introduction. The method part is informative. The results are detailed in an convincing manner. The discussion concludes the limitations and is meaningful.
I found aspects which might be discussed in the manuscript.
- Are there any guidelines on PEP after rape? Is rape itself an indication for PEP?
- Is Belgium a low or high prevalence country for HIV? How does the influence recommendation or completion rate for PEP after rape? This should be part of the introduction and the discussion.
Author Response
We sincerely thank you for your thorough review of our manuscript entitled "The role of centralized sexual assault care centers in HIV post exposure prophylaxis treatment adherence. A retrospective single center analysis." We appreciate the insightful comments and constructive suggestions, which have helped us improve the quality and clarity of our work.
Below, we provide a detailed point-by-point response to each of the reviewers’ comments. Reviewer comments are presented in bold, followed by our responses in regular text with the modification to the manuscript highlighted in italic. All changes made in the manuscript are highlighted accordingly.
I found aspects which might be discussed in the manuscript.
- Are there any guidelines on PEP after rape? Is rape itself an indication for PEP?
We thank you for the comment.
Indeed national guidelines exist that are not endorsed by the federal health agency. We provided the information and reference in the introduction and specified that within these guidelines rape itself is not an indication for PEP with treatment formally recommended only for anal and vaginal intercourse. We modified as following the introduction:
In Belgium, HIV PEP for sexual assault victims has traditionally been administered in emergency departments (EDs), guided by national guidelines to decide on treatment initiation according to types of exposure and delivered by personnel often lacking specialized training in sexual assault care.
- Is Belgium a low or high prevalence country for HIV? How does the influence recommendation or completion rate for PEP after rape? This should be part of the introduction and the discussion.
We thank you for the suggestion as indeed HIV prevalence is an important factor that has to be taken into account when interpreting our results. As suggested we added the following sentence to the introduction:
Belgium is a low prevalence country with prevalence, outside high-risk groups is estimated below 0.02%.
Moreover in the introduction we modified a sentence as follows to highlight that PEP is not indicated per se after sexual assault but needs to be evaluated according to the exposure type:
In Belgium, HIV PEP for sexual assault victims has traditionally been administered in emergency departments (EDs), guided by national guidelines to decide on treatment initiation according to the exposure and delivered by personnel often lacking specialized training in sexual assault care.
Moreover we added the following sentence in the discussion to acknowledge that completion rates might have been influenced by the ‘a priori’ low HIV prevalence in our context:
The low HIV prevalence within our context might have influenced PEP completion rates.
We hope that the revisions and clarifications provided address all concerns raised. We are grateful for the opportunity to revise our manuscript and look forward to your feedback.
Round 2
Reviewer 1 Report
Comments and Suggestions for Authors
It is recommended to reconstruct the conclusion by clearly stating the necessity for improving HIV PEP outcomes. This could be significant for providing key insights to policymakers and clinicians involved in crafting nPEP care.
Author Response
It is recommended to reconstruct the conclusion by clearly stating the necessity for improving HIV PEP outcomes. This could be significant for providing key insights to policymakers and clinicians involved in crafting nPEP care.
We thanks the reviewer for his/her comment on our work.
We changed our conclusion in the abstract as follows:
SAC-centered care is not associated with an increase in PEP completion rates in sexual assault victims beyond the increase associated with improved access to early and delayed psychological support. Others measures to improve PEP completion rates should be developed.
In the manuscript conclusion we reconstructed the wording in order to comply with the Reviewer suggestions as follows:
In conclusion, an holistic PEP program for sexual assault victims should focus on adherence to the prescribed regimen and interventions improving PEP outcomes. Our study shows that instauration of a SAC does not improve completion rates of the prescribed 28-days PEP regimen while it improves access to early psychological support, which in turn increases treatment adherence.
We hope that the revisions and clarifications provided address all concerns raised. We are grateful for the opportunity to revise our manuscript and look forward to your feedback.
Sincerely,
Stefano Malinverni, on behalf of all co-authors
